Automatic single- and multi-label enzymatic function prediction by machine learning

Amidi Shervine 1
Amidi Afshine 1
http://orcid.org/0000-0003-1823-6102 Vlachakis Dimitrios 2
Paragios Nikos 1 3
Zacharaki Evangelia I. 1 3 evangelia.zacharaki@ecp.fr
1 Department of Applied Mathematics, Center for Visual Computing, Ecole Centrale de Paris (CentraleSupélec) , Châtenay-Malabry , France
2 MDAKM Group, Department of Computer Engineering and Informatics, University of Patras , Patras , Greece
3 Equipe GALEN, INRIA Saclay , Orsay , France
Valencia Alfonso
Electronic publication date: 2017 Mar 29
Publication date: 2017
Volume: 5
Electronic Location ID: e3095
Received 2016 Nov 30; Accepted 2017 Feb 15
Copyright: © 2017 Amidi et al.
Copyright year: 2017
Copyright holder: Amidi et al.
License: This is an open access article distributed under the terms of the Creative Commons Attribution License, which permits unrestricted use, distribution, reproduction and adaptation in any medium and for any purpose provided that it is properly attributed. For attribution, the original author(s), title, publication source (PeerJ) and either DOI or URL of the article must be cited.
License URL: https://creativecommons.org/licenses/by/4.0/

Keywords: Enzyme classification, Single-label, Multi-label, Structural information, Amino acid sequence, Smith-Waterman algorithm

Funding: European Research Council Grant Diocles ERC-STG-259112 This research was partially supported by European Research Council Grant Diocles (ERC-STG-259112). There was no additional external funding received for this study. The funders had no role in study design, data collection and analysis, decision to publish, or preparation of the manuscript.

==============================
The number of protein structures in the PDB database has been increasing more than 15-fold since 1999. The creation of computational models predicting enzymatic function is of major importance since such models provide the means to better understand the behavior of newly discovered enzymes when catalyzing chemical reactions. Until now, single-label classification has been widely performed for predicting enzymatic function limiting the application to enzymes performing unique reactions and introducing errors when multi-functional enzymes are examined. Indeed, some enzymes may be performing different reactions and can hence be directly associated with multiple enzymatic functions. In the present work, we propose a multi-label enzymatic function classification scheme that combines structural and amino acid sequence information. We investigate two fusion approaches (in the feature level and decision level) and assess the methodology for general enzymatic function prediction indicated by the first digit of the enzyme commission (EC) code (six main classes) on 40,034 enzymes from the PDB database. The proposed single-label and multi-label models predict correctly the actual functional activities in 97.8% and 95.5% (based on Hamming-loss) of the cases, respectively. Also the multi-label model predicts all possible enzymatic reactions in 85.4% of the multi-labeled enzymes when the number of reactions is unknown. Code and datasets are available at https://figshare.com/s/a63e0bafa9b71fc7cbd7.

Introduction

The ever-growing PDB database contains more than 110,000 proteins that are characterized by different properties including their structure, biological function, chemical composition, or solubility in solvents. Protein classification is important since it allows estimating the properties of novel proteins according to the group to which they are predicted to belong. Enzymes are a type of proteins that are classified according to the chemical reactions they catalyze into six primary classes, oxidoreductases, transferases, hydrolases, lyases, isomerases, and ligases. The classes are denoted by the enzyme commission (EC) number (NC-IUBMB, 1992) and have been determined based on experimental evidence. Systematic annotation, reliability, and reproducibility of protein functions are discussed in Valencia (2005). Classification of enzymes is a central issue because it helps understanding enzymatic behavior during chemical reactions. While the vast majority of enzymes have been found to perform particular reactions, a non-negligible number of enzymes can perform different reactions and can hence be directly associated with multiple enzymatic functions (Guyon et al., 2006).

During the last decade, various machine learning techniques have been proposed for both single-label and multi-label enzyme classification on different datasets. Among single-label classification studies, some (Dobson & Doig, 2005) used only structural information and achieved an accuracy of 35% for top-ranked prediction using support vector machine (SVM) with a one-against-one voting scheme on 498 enzymes from the PDB database. Applying SVM on sequence features has been done by Mohammed & Guda (2015) and achieved an accuracy of 98.39% after training on 150,000+ enzymes with 10-fold cross-validation. Osman & Choong-Yeun Liong (2010) extracted only gene or amino acid sequence information and applied neural networks obtaining an accuracy of 72.94% after training the networks on 1,200 enzymes from the PDB database and testing on 2,000 others. Volpato, Adelfio & Pollastri (2013) achieved 96% accuracy with a 10-fold cross-validation scheme on 6,081 entries of the ENZYME database. Sequence structure and amino acid information were also used by des Jardins et al. (1997), Kumar & Choudhary (2012) and Lee et al. (2007), who obtained testing accuracies ranging from 74% to 88.2% using the Swiss-Prot database. Combination of sequence, structure, and chemical properties of enzymes was also explored by Borgwardt et al. (2005) using kernel methods and SVM on the BRENDA database and achieved an accuracy of 93% with six-fold cross-validation on information extracted through protein graph models. Multi-label classification using different methods such as RAkEL-RF and MLKNN (Wang et al., 2014) or MULAN (Zou et al., 2013) was performed on single- and multi-labeled enzymes. In particular, the latter was assessed on enzymes from the Swiss-Prot database based on their amino acid composition and their physico-chemical properties and involved the use of position-specific scoring matrices. In the best scenario, a macro-averaged precision of 99.31% was obtained on a set of 2,840 multi-functional enzymes after 10-fold cross-validation. A summary of other alignment-free methods used to predict enzyme classes is presented in Table 1.

Table 1 Comparative table of several alignment-free approaches.

No. proteins	Information	Parameters	Classification method	Level	Work	
1,371	3D structure	3D-HINT potential	LDA	QSAR	ANN	0–1	Concu et al. (2009c)	
4,755	Moments, entropy, electrostatic, HINT potential	MLP	Concu et al. (2009b)	
2,276	3D-QSAR	Concu et al. (2009a)	
26,632	Global binding descriptors	SVM	1–3	Volkamer et al. (2013)	
211,658	Structural	GRAVY	1	Dave & Panchal (2013)	
3,095	Sequence	PseAAC, SAAC, GM	ML-kNN	Zou & Xiao (2016)	
9,832	FunD, PSSM	OET-kNN	1–2	Shen & Chou (2007)	
300,747	Interpro signatures	BR-kNN	1–4	Ferrari et al. (2012)	

Other work on enzyme classification includes the use of information stemming from topological indices (Munteanu, Gonzalez-Diaz & Magalhaes, 2008), peptide graphs (Concu et al., 2009b), and also includes the machine-learning based ECemble method (Mohammed & Guda, 2015).

In this paper, a new feature extraction and classification scheme is presented that combines both structural and amino acid sequence information aiming to improve standard classifiers that use only one type of information. Building upon previous work (Amidi et al., 2016), we investigate a more sophisticated combination approach and assess the performance of the scheme in single-label and multi-label classification tasks. State-of-the-art accuracy is observed as compared to the methods reviewed in the survey by Yadav & Tiwari (2015).

Methods

Feature extraction

Proteins are chains of amino acids joined together by peptide bonds. As the three-dimensional (3D) configuration of the amino acids chain is a very good predictor of protein function, there has been many efforts in extracting an appropriate representation of the 3D structure (Lie & Koehl, 2014). Since many conformations of this chain are possible due to the possible rotation of the peptide bond planes relative to each other, the use of rotation invariant features is preferred over features based on Cartesian coordinates of the atoms. In this study, the two torsion angles of the polypeptide chain were used as structural features. The two torsion angles describe the rotation of the polypeptide backbone around the bonds between N − Cα (angle φ) and Cα − C (angle ψ). The probability density of the torsion angles φ and ψ ∈ [−180°, 180°] was estimated by calculating the 2D sample histogram of the angles of all residues in the protein. When the protein consisted of more than one chain, the torsion angles of all chains were included together into the feature vector. Smoothness in the density function was achieved by moving average filtering, i.e., by convoluting the 2D histogram with a uniform kernel. The range of angles was discretized using 19 × 19 bins centered at 0° and the obtained matrix of structural features was linearized to a 361-dimensional feature vector for each enzyme representing structural information (XSI).

Although structure relates to amino acid sequence, additional information can be extracted directly from the protein sequences. Assessment of similarities between amino acid sequences of enzymes is usually performed by sequence alignment. The Smith–Waterman sequence alignment algorithm (Smith & Waterman, 1981) has been preferred over the Needleman–Wunsch algorithm (Needleman & Wunsch, 1970) due to the assessment of sequence similarity based on local alignment (in contrast to the global alignment previously performed), which enables possible deletions, insertions, substitutions, matches and mismatches of arbitrary lengths. Optimizing local alignment allows to take into consideration mutations that might have happened in amino acid sequences. The similarity of each pair of sequences i and j can be quantified using the scoring matrix that is produced by the sequence alignment algorithm. For two sequences i and j, the highest score in the previous matrix, which reflects the success of alignment, is used as similarity criterion S(i, j). Amino acid sequence information is represented in two distinct ways. First, for each one of the six classes, the similarity matrix S of a sequence to all training samples of that class is calculated and summarized as a histogram vector with 10 bins. The six-histogram vectors are then concatenated into a 60-dimensional feature vector which is denoted as XAA. Second, the class probabilities (FAA)xj of a given enzyme j are expressed as the maximum similarities S within each class normalized over all classes:  For each class x,  (fAA)xj=maxk∈ training ∩ EC xk≠jS(k,j)∑l=16maxk∈ training ∩ EC lk≠jS(k,j)

Classification and fusion

Two classification techniques have been investigated, the nearest neighbor (NN) and SVM. NN is preferred for its simplicity and its small computation time whereas SVM is useful to find non-linear separation boundaries. The classifiers are trained using a number of annotated examples and then tested on novel enzymes. Two types of classification models have been produced: single-label models for the enzymes performing unique reactions and multi-label models for the multi-functional enzymes. Both structural (SI) and amino acid sequence (AA) information is related to the enzymatic activity. In order to take into consideration these two properties, fusion of information is performed in two different ways, in the feature level and in the decision level.

The concept of the feature-level fusion is to concatenate the two sets of 361 structural and 60 amino acid sequence features before performing classification. The feature-level fusion approach is illustrated in Fig. 1.

Figure 1 Overview of feature-level fusion.

The decision-level fusion approach associates class probabilities for SI obtained by SVM (Platt, 1999) fSISVM or NN (Atiya, 2005) fSINN with class probabilities for AA (fAA) through a heuristic fusion rule. The applied fusion rule performs weighted averaging of class probabilities using unequal weights. Thus, the corresponding fused class probability is given by (1 − α)(fSI) + α(fAA). An optimized α is empirically obtained for each classification method by maximizing the accuracy over the training data (Amidi et al., 2016). A single class is assigned in the single-label classification (Fig. 2) based on the maximum probability.

Figure 2 Decision-level fusion for single- and multi-label classification.

This hard decision rule cannot be applied to the multi-label scenario. In order to obtain a soft decision, a multi-label classifier is applied on the fused class probabilities to produce the final decision outputs. In particular, the six-dimensional class probabilities (fused from AA and SI) are introduced into a multi-label SVM or multi-label NN, which computes a six-dimensional binary vector where the cth feature is equal to 1 if the predicted enzyme belongs to class c and 0 otherwise.

Performance assessment

The data have been randomly split into 80% for training and validation and 20% for independent testing. The training/validation set has been divided into five random folds that have been used to determine the optimal parameters. More particularly, the parameters were optimized by a standard five-fold cross-validation based on classification accuracy in the single-label classification problem and on the subset accuracy in the multi-label classification problem. Upon optimization, the parameters were fixed and remained the same throughout all experiments. Then for both classification problems, performance has been assessed by applying the methods on the independent testing set using the fixed parameters.

Single-label classification

The performance of single-label classification has been assessed based on the confusion matrix whose elements C(x, y) with x, y ∈ ⟦1, 6⟧, indicate the number of enzymes that belong to class x and are predicted as belonging to class y. Two metrics are based on this definition: the overall accuracy that evaluates the proportion of correctly classified enzymes among the total number of enzymes and the balanced accuracy that avoids inflated performance estimates on imbalanced datasets. They are defined by: Overall Accuracy=∑x=16C(x,x)∑x,y=16C(x,y)  and  Balanced Accuracy=16⋅∑x=16C(x,x)∑y=16C(x,y)

Multi-label classification

In the case of multi-label classification, the labels of an enzyme i are represented by a six-dimensional binary vector Li where the value 1 at a position j ∈ ⟦1, 6⟧ indicates the positivity of class j and 0 otherwise. Also, we denote with N the total number of enzymes, as well as Litrue and Lipred the sets of true and predicted labels of enzyme i, respectively. The performance of the multi-label classifiers cannot be assessed using the exact same definitions as for the single-label classifiers. Various multi-label metrics defined in previous works (Zhang & Zhou, 2006; Tsoumakas & Katakis, 2007; Madjarov et al., 2012) have been considered in our study. Here, we introduce the Kronecker delta δ, the symmetric difference Δ, the binary union ∪ and intersection ∩ operations, as well as the l1-norm | |. The following metrics have been chosen to assess the performance of our new method: Hamming-loss assesses the frequency of misclassification of a classifier on a given set of enzymes. This index is averaged over all classes and all enzymes. Also, we will note 1-Hamming-loss the complementary of this indicator so that the worst-case value is 0 and the best is 1. Conversely, to the Hamming-loss index, the latter assesses the average over all enzymes of the proportion of binary class memberships that are correctly predicted. Hamming-Loss=1N∑i=1N16|LipredΔLitrue|

Accuracy averages over all enzymes the Jaccard similarity coefficient of the predicted and true sets of labels. This index reflects the averaged proportion of similar class membership between those two sets. Accuracy=1N∑i=1N|Lipred∩Litrue||Lipred∪Litrue|

Precision, recall, and F1 score, which have been adapted for multi-label classification. The two first metrics respectively reflect the proportion of detected positives that are effectively positive, and the proportion of positives samples that are correctly detected. Finally, the F1 score balances the information provided by these two indexes through the computation of an harmonic mean. Precision=1N∑i=1N|Lipred∩Litrue||Lipred| Recall=1N∑i=1N|Lipred∩Litrue||Litrue| F1=2N∑i=1N|Lipred∩Litrue||Lipred|+|Litrue|

Subset accuracy considers that a given enzyme is correctly classified if and only if all class memberships are correctly predicted. This metric is the strictest of this study, since it requires the sets of true and predicted labels to be identical in order for an enzyme to be considered as correctly classified.

Subset accuracy=1N∑i=1Nδ(Lipred,Litrue)

Macro-precision, recall and F1 compute, respectively precision, recall, and F1-score separately for each class, and then average the values over the six classes. These indexes are crucial for us, as they highlight the performance of our method on small-populated labels. In the following definitions, TPj and FPj, respectively represent the number of true positives and false positives, and Precisionj and Recallj are those associated to label j ∈ ⟦1, 6⟧, considered as binary class.

M-precision=16∑j=16TPjTPj+FPj M-recall=16∑j=16TPjTPj+FNj M-F1=26∑j=16Precisionj×RecalljPrecisionj+Recallj

Micro-precision, recall, and F1 are similar to the single-label definition of those three quantities, whereas here they rely on the values of the sum over all classes of true positives, false positives, and false negatives. The micro indexes indicate whether the majority of the enzymes are correctly classified, regardless if they belong to low- or high-populated classes.

m-precision=∑j=16TPj∑j=16TPj+∑j=16FPj m-recall=∑j=16TPj∑j=16TPj+∑j=16FNj m-F1=2⋅m-precision×m-recallm-precision+m-recall

Data

The method has been applied on data from the PDB database that include one set of single-labeled enzymes (Table 2) and one set of multi-labeled enzymes (Table 3).

Table 2 Dataset I: 39,251 single-labeled enzymes.

Class	EC 1	EC 2	EC 3	EC 4	EC 5	EC 6	
Name	Oxidoreductase	Transferase	Hydrolase	Lyase	Isomerase	Ligase	
Number	7,256	10,665	15,451	2,694	1,642	1,543	

Table 3 Dataset II: 783 multi-labeled enzymes.

Number of classes	2	3	4	
EC numbers	1	1	1	1	2	2	2	2	3	3	3	4	1	1	1	1	
2	3	4	5	3	4	5	6	4	5	6	5	2	2	4	2	
												3	4	5	4	
														5		
Number of enzymes	62	44	14	2	217	160	45	15	82	23	73	28	1	7	6	4	
Note:

The total number of enzymes with 2, 3 and 4 labels each are 765, 14, 4, respectively.

Results

Single-label classification

Classification via decision-level fusion has been performed using α = 0.95 for the SVM method and α = 0.99 for the NN method. Overall and balanced accuracies obtained with each method on the testing set are detailed in Table 4.

Table 4 Testing performance of dataset I.

Type	SI	AA	Decision fusion	Feature fusion	
Classifier	SVM	NN	NN	SVM	NN	SVM	NN	
Overall accuracy	0.830	0.828	0.976	0.977	0.978	0.942	0.878	
Balanced accuracy	0.755	0.788	0.968	0.966	0.968	0.910	0.856	

The decision-level fusion classification increased the overall accuracy by 0.2% compared to the best results obtained by either AA only or SI only. The balanced accuracy achieved is 96.8%, which is the same as the one achieved by NN classification using AA only. Also, SVM classification via feature-level fusion achieves 11.2% higher overall accuracy than classification via SI only but 3.4% less overall accuracy than NN classification on AA only. In general, classification using SVM tends to achieve better overall accuracy than with NN (0.2% and 6.4%, respectively for SI only and feature-level fusion), whereas except for the feature-level fusion, NN tends to achieve better balanced accuracy than SVM (0.2% and 3.3%, respectively for decision-level fusion and SI only).

Multi-label classification

As described in the methods’ section, the optimal fusion parameter α was empirically determined for each dataset (single- or multi-functional) and fusion scheme. The optimal values are shown in Fig. 3 from which it can be seen that the values of α for the decision-level fusion in multi-label classification (α = 0.69, 0.73, 0.76, 0.80, respectively for the SVM–NN, SVM–SVM, NN–NN, and NN–SVM methods) are approximately 20% smaller compared to the values obtained in single-label classification (α = 0.95, 0.99, respectively for the SVM and NN methods). This shows that structural information plays a more significant role in differentiating enzymatic activity in the case of multi-labeled enzymes than in single-label classification (which is mostly based on amino-acid sequence information).

Figure 3 Testing subset accuracy for dataset II.

Figure 3 shows the subset accuracy for the testing set obtained for each approach in multi-label classification. Both SVM and NN classifiers achieved approximately 10% less subset accuracy when using only SI than when using only AA. Combining SI and AA according to the feature-level fusion scheme leads to intermediate values (between the ones achieved by only SI and AA) of subset accuracy. However, the combination of information based on the decision-level fusion scheme increased the subset accuracy by up to 1.3% compared to the best approach using AA only. The best results were obtained with the SVM–NN classification scheme. The overlap and discrepancy in correct predictions using SI (SVM), AA (NN) and the decision-level fusion scheme with SVM–NN are illustrated in Fig. 4.

Figure 4 Repartition of correctly predicted enzymes with respect to subset accuracy.

We observed that 65.6% of the enzymes in the testing test were correctly predicted by all compared approaches (SI only, AA only, and decision-level fusion). Also, out of 29 enzymes correctly predicted by AA but not by SI, 28 are also correctly predicted by the SVM–NN decision-level fusion scheme. This shows that the decision-level fusion incorporates the relevant information provided by AA, which was missed by SI. Conversely, out of five enzymes correctly predicted by SI and not by AA, two of them are correctly predicted by the decision-level fusion scheme. This could be related to the chosen values of α that assigns a larger weight to the class probabilities calculated by AA than the ones extracted from SI.

Computation of 1-Hamming-loss for each approach is shown in Fig. 5. All decision-level fusion schemes achieved higher values than the approaches using only AA or only SI. The decision-level scheme that performed best in terms of Hamming-loss is SVM–SVM with an increase of 1.8% compared to AA (NN). The comparison of 1-Hamming-loss per class for each best method (SI only, AA only and decision-level fusion) is shown in Table 5.

Figure 5 Testing 1-Hamming-loss for dataset II.

Table 5 Comparison of 1-Hamming-loss per class with SVM–SVM.

Classifier	1-Hamming-loss per class	
EC 1	EC 2	EC 3	EC 4	EC 5	EC 6	
SI SVM only	0.962	0.834	0.860	0.822	0.943	0.962	
AA NN only	0.962	0.930	0.898	0.885	0.962	0.987	
Decision fusion SVM–SVM	0.968	0.917	0.949	0.943	0.968	0.987	
Note:

The best classification performance is indicated in bold for each class.

The SVM–SVM method achieves for each class except for the transferases up to 5.8% higher 1-Hamming-loss than the maximum accuracy achieved by the best classifier of a single type of information (SI or AA). There is an increase in the performance regardless of the size of the class. In particular, classification of a large class such as the hydrolases had a 5.1% increase in 1-Hamming-loss, whereas small classes like the lyases and isomerases were classified respectively with 5.8% and 0.6% better performance after fusion than with SI or AA only.

Table 6 shows the results of the 10 methods, according to all of the metrics that have been assessed for multi-label classification. With respect to all the indexes, we observe that the decision-level fusion schemes outperform those carrying only one type of information. More particularly, each of the SVM–SVM, SVM–NN, and NN–SVM techniques provide a distinct advantage in the process of multi-label classification. First of all, the SVM–SVM scheme is best in terms of 1-Hamming-loss with a testing value of 95.5%, which surpasses other methods by at least 1% margin. Also, this scheme proves to be the best in terms of the three definitions of recall, meaning that if an enzyme belongs to a certain class, SVM–SVM will be the more likely to detect it. In terms of predicting exact matches of the true labels, the SVM–NN method will be the best one to consider with a testing value of 85.4%, which is at least 1.3% ahead of the performance achieved considering only one type of information. One of the most impressive rises in performance stems from the NN–SVM method, which proves to outperform SI and AA methods by +4.1%, +2.6%, and +4.6% in terms of precision, M-precision and m-precision, respectively. Not only does it show that the relevance of class predictions is improved overall, but also and more importantly that small-populated classes benefit from this progression as well.

Table 6 Testing performance of dataset II.

Type	SI	AA	Decision fusion	Feature fusion	
Classifier	SVM	NN	SVM	NN	SVM	NN	SVM	NN	
SVM	NN	SVM	NN	
Alpha					0.73	0.69	0.80	0.76			
Hamming-loss	0.103	0.119	0.064	0.063	0.045	0.054	0.054	0.063	0.083	0.098	
Accuracy	0.790	0.800	0.883	0.885	0.906	0.898	0.879	0.889	0.823	0.831	
Precision	0.857	0.829	0.901	0.906	0.942	0.918	0.947	0.907	0.889	0.856	
Recall	0.825	0.831	0.908	0.908	0.924	0.920	0.885	0.911	0.847	0.856	
F1 score	0.835	0.829	0.904	0.906	0.928	0.919	0.893	0.908	0.859	0.855	
Subset accuracy	0.688	0.739	0.834	0.841	0.847	0.854	0.841	0.847	0.726	0.783	
Macro	Precision	0.921	0.744	0.940	0.941	0.962	0.945	0.967	0.903	0.927	0.806	
Recall	0.741	0.777	0.881	0.871	0.887	0.879	0.854	0.881	0.791	0.787	
F1	0.801	0.758	0.902	0.897	0.921	0.905	0.905	0.889	0.844	0.794	
Micro	Precision	0.864	0.822	0.904	0.907	0.943	0.919	0.953	0.904	0.901	0.857	
Recall	0.829	0.832	0.910	0.910	0.925	0.922	0.885	0.913	0.850	0.857	
F1	0.846	0.827	0.907	0.908	0.934	0.921	0.918	0.909	0.875	0.857	
Note:

The best classification performance (based on different criteria) is indicated in bold for each technique.

The code was written in Matlab and Python languages and is freely and publicly available at https://figshare.com/s/a63e0bafa9b71fc7cbd7. Running on a single Intel Xeon X5650 processor, the average prediction time of the enzymatic function(s) of a new enzyme was less than 3 s. Computations were achieved using high performance computing (HPC) resources from the “mesocentre” computing center of Ecole Centrale de Paris (http://www.mesocentre.ecp.fr) supported by CNRS.

Discussion and Conclusion

The results of both single-label and multi-label classifications showed that the combination of information leads to more accurate enzyme class prediction than the individual structural or amino acid descriptors. Among fusion approaches, the decision-level fusion performed better than the feature-level fusion. In the multi-label case, the SVM–NN fusion scheme achieved the best subset accuracy by predicting correctly the labels of 85.4% of the enzymes. The NN–NN fusion scheme also performed well (84.7%) and required the least computational time during the training phase. Structural information seems to be more important in the case of multi-label classification than in single-label, since the optimal relative weight of amino acid sequence features during fusion was found to be smaller in multi-labeled enzymes (α ∈ [0.69, 0.80]) compared to single-labeled enzymes (α ∈ [0.95, 0.99]).

In all examined cases, AA was more informative than SI in respect to the prediction of enzymatic activity. The same trend has been observed in a study of Zou et al. (2013) which showed an increase of 0.81% with sequence related features, compared to structural features. However, it should be noted that we examined only general functional characteristics indicated by the first digit of EC code. A study assessing the relationship between function and structure (Todd, Orengo & Thornton, 2001) revealed 95% conservation of the fourth EC digit for proteins with up to 30% sequence identity. Similarity, Devos & Valencia (2000) concluded that enzymatic function is mostly conserved for the first digit of EC code whereas more detailed functional characteristics are poorly conserved.

The single- and multi-label classification models have been trained and tested on enzymes assumed to perform single or multiple reactions, correspondingly. However, the single-label enzymes might be associated with other reactions not detected yet and in fact be multi-label. In order to assess the method in a more general scenario, we mixed both single- and multi-label information during training phase and observed a slight improvement in prediction accuracy. Specifically, we chose to examine the NN–NN fusion scheme because of its small computation time, and merged SI and AA probabilities obtained by both datasets I and II. This model achieved 89.2% subset accuracy and 95.8% accuracy for the multi-label dataset (by cross-validation) indicating an increase of 4.5% and 2.1% in respect to the results obtained with the NN–NN scheme trained only on multi-labeled data (shown in Figs. 3 and 5). This also corresponds to an increase of 3.8% and 0.3%, respectively, from the best fusion schemes.

Moreover, since it is unknown for new (testing) enzymes if they perform unique reactions, they have to be treated as multi-label. In order to estimate the performance of the single-label model in the case of unknown enzymes, we tested the best single-label classifier (i.e., the NN on the decision level) on the multi-label dataset. For 93.0% of the enzymes the model predicted correctly one of their actual labels, whereas the prediction of all actual labels (by selecting the classes with the highest probability scores) was correct in 44.8% of the enzymes.

Furthermore, we investigated techniques dealing with imbalanced classes but did not observe any conclusive outcome. In particular, ADASYN improved overall accuracy on the single-label dataset by 0.1% but reduced balanced accuracy by 1.1%.

In conclusion, computational models calculated from experimentally acquired annotations of large datasets provide the means for fast, automated, and reproducible prediction of functional activity of newly discovered enzymes and thus can guide scientists in deciphering metabolic pathways and in developing potent molecular agents. Future work includes the representation of the whole 3D geometry using additional structural attributes and the incorporation of deep learning architectures that have proven to be powerful tools in supervised learning. The main advantage of deep learning techniques is the automatic exploitation of features and tuning of performance in a seamless fashion, that optimizes conventional analysis frameworks.

The authors wish to thank Prof. V. Megalooikonomou from the MDAKM group, Department of Computer Engineering and Informatics, University of Patras, for his earlier collaboration on structural similarity. The authors would also like to thank Chloé-Agathe Azencott for useful discussion about the Smith–Waterman algorithm.

Additional Information and Declarations

Competing Interests

Author Contributions

Data Deposition

Nikos Paragios and Evangelia Zacharaki are employees of Equipe GALEN, INRIA Saclay, France.

Shervine Amidi conceived and designed the experiments, performed the experiments, analyzed the data, contributed reagents/materials/analysis tools, wrote the paper, prepared figures and/or tables, and reviewed drafts of the paper.

Afshine Amidi conceived and designed the experiments, performed the experiments, analyzed the data, contributed reagents/materials/analysis tools, wrote the paper, prepared figures and/or tables, and reviewed drafts of the paper.

Dimitrios Vlachakis conceived and designed the experiments, contributed reagents/materials/analysis tools, wrote the paper, and reviewed drafts of the paper.

Nikos Paragios conceived and designed the experiments, wrote the paper, and reviewed drafts of the paper.

Evangelia I. Zacharaki conceived and designed the experiments, performed the experiments, analyzed the data, wrote the paper, and reviewed drafts of the paper.

The following information was supplied regarding data availability:

Amidi, S. (2017): Code. figshare. DOI 10.6084/m9.figshare.4621672.v1.

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
