# Peer review of "Automatic single- and multi-label enzymatic function prediction by machine learning"

_PeerJ, doi:10.7717/peerj.3095_

## Round 0.1 · original submission · Major Revisions

· Academic Editor

Major Revisions

Dear Authors,

As you can see the three referees have a number of comments and requests that will make the paper stronger and easier to follow.
In summary, they ask for a more comprehensive review of a field in which a substantial number of papers have been published. They suggest some papers and there are many others.

They request details on the methodology and a more clear evaluation of the results.

Finally, the referees suggest (and I agree very much) that the deposition of the datasets and software will make the results reproducible by others and increase its impact in the community.

Reviewer 1 ·

Basic reporting

The authors are presenting a methodology that is mixing peptide torsion angle and sequence alignment information to predict single and multi-label enzymatic function.
In the Introduction section, the authors should mention other information coding that have been used for enzyme classification such as topological indices of peptide graphs [1,2].
About the data used for the study: it will be very useful if the raw dataset will be available in any open data repository (FigShare, etc.).

References
[1] Enzymes/non-enzymes classification model complexity based on composition, sequence, 3D and topological indices, Journal of Theoretical Biology 254(2), 476-482 (2008)
[2] 3D Entropy & Moments Prediction of Enzyme Classes and Experimental-Theoretic Study of Peptide Fingerprints in Leishmania Parasites, Biochimica et Biophysica Acta (BBA) - Proteins & Proteomics 1794(12), 1784–1794 (2009)

Experimental design

The authors should explain why they used a new methodology to obtain these classifications during other publications presented a multi-label classification with 99.54 accuracy (line 38). How the current results are improving this number?
It should be useful to present details when the authors are comparing results with previous ones: number of features, number of folds, type of data set split, etc.
The reproducibility of these results should be improved by placing the raw dataset and the Matlab / python scripts as open repositories (GitHub, FigShare, etc.). Any scientist should be able to download the dataset and to run the script.
It should be explained the reason for choosing the current type of data split and number of folds.
When method performance are presented, it should be mentioned if there are training or test values.

Validity of the findings

It should be explained better the advantage of this method compared with the other ones: using previous 99.54 accuracy, execution times, model complexity, etc.
It should be specify that the authors are expecting for the deep learning methods by including new information.
Why the authors are not using peptide graphs descriptors as invariant codification? (ex: protein star-graphs, protein contact networks using amino acid properties, etc.).

Additional comments

The manuscript is presenting an interesting method to mix molecular information for this type of classification of enzymes.
The main three weak spots: the argument for using a new classification comparing with the previous 99.54 accuracy, why there is no molecular graph information involved and the reproducibility of the results without open dataset and scripts.
The details about each point are presented in the above review sections.

Reviewer 2 ·

Basic reporting

no comments

Experimental design

no comments

Validity of the findings

no comments

Additional comments

I recommend to insert a short section to compare this method with other alignment-free methods used to predict enzyme classes or even second level enzyme sub-classes. In this sense, I recommend to insert an discuss briefly a summary comparative table including items like: number of sequences studied, type of structural information considered (sequence, 3D structure), Machine learning methods used (ANN, LDA, etc), type of structural parameters (entropies, moments, etc.), number of classes or sub-classes of enzymes predicted. Some of the previous works you may consider to discuss and cite in this area are the following:
1: Zou HL, Xiao X. Classifying Multifunctional Enzymes by Incorporating Three
Different Models into Chou's General Pseudo Amino Acid Composition. J Membr Biol.
2016 Aug;249(4):551-7. doi: 10.1007/s00232-016-9904-3. PubMed PMID: 27113936.


2: Dave K, Panchal H. ENZPRED-enzymatic protein class predicting by machine
learning. Curr Top Med Chem. 2013;13(14):1674-80. Review. PubMed PMID: 23889047.


3: Volkamer A, Kuhn D, Rippmann F, Rarey M. Predicting enzymatic function from
global binding site descriptors. Proteins. 2013 Mar;81(3):479-89. doi:
10.1002/prot.24205. PubMed PMID: 23150100.


4: De Ferrari L, Aitken S, van Hemert J, Goryanin I. EnzML: multi-label
prediction of enzyme classes using InterPro signatures. BMC Bioinformatics. 2012
Apr 25;13:61. doi: 10.1186/1471-2105-13-61. PubMed PMID: 22533924; PubMed Central
PMCID: PMC3483700.

5. Shen HB, Chou KC. EzyPred: a top-down approach for predicting enzyme
functional classes and subclasses. Biochem Biophys Res Commun. 2007 Dec
7;364(1):53-9. PubMed PMID: 17931599.


1: Concu R, Dea-Ayuela MA, Perez-Montoto LG, Prado-Prado FJ, Uriarte E,
Bolás-Fernández F, Podda G, Pazos A, Munteanu CR, Ubeira FM, González-Díaz H. 3D
entropy and moments prediction of enzyme classes and experimental-theoretic study
of peptide fingerprints in Leishmania parasites. Biochim Biophys Acta. 2009
Dec;1794(12):1784-94. doi: 10.1016/j.bbapap.2009.08.020. PubMed PMID: 19716935.


2: Concu R, Dea-Ayuela MA, Perez-Montoto LG, Bolas-Fernández F, Prado-Prado FJ,
Podda G, Uriarte E, Ubeira FM, González-Díaz H. Prediction of enzyme classes from
3D structure: a general model and examples of experimental-theoretic scoring of
peptide mass fingerprints of Leishmania proteins. J Proteome Res. 2009
Sep;8(9):4372-82. doi: 10.1021/pr9003163. PubMed PMID: 19603824.


3: Concu R, Podda G, Uriarte E, González-Díaz H. Computational chemistry study of
3D-structure-function relationships for enzymes based on Markov models for
protein electrostatic, HINT, and van der Waals potentials. J Comput Chem. 2009
Jul 15;30(9):1510-20. doi: 10.1002/jcc.21170. PubMed PMID: 19086060.

Reviewer 3 ·

Basic reporting

The paper is well written however, I may suggest authors change "increased" with "have been increasing" in the line 1 of the abstract.
The Introduction between line 24 and 38 is short and the authors should consider also the papers aimed at the prediction of the first EC number using a 3D approach.

Experimental design

No Comment

Validity of the findings

The method used by the authors seems to be robust and consistent. However, I think authos should provide more data supporting the robustness of the approach like the ROC curve, the Matthews correlation coefficient, etc.
In addition, I think the authors should further discuss the results and the significance of the features in the prediction models. Moreover, it is not clear for me if the authors applied any kind of feature selection or the models will include all the features they calculated.

---

## Round 0.2 · accepted · Accept

· Academic Editor

Accept

Dear Authors,

Thanks for revising the manuscript. The new version contains the details about the methods, new references and availability of the software that were previously required.